# Targeted Approaches to HER2-Low Breast Cancer: Current Practice and Future Directions

**DOI:** 10.3390/cancers14153774

**Published:** 2022-08-03

**Authors:** Heng-Zhou Lai, Jie-Rong Han, Xi Fu, Yi-Feng Ren, Zhuo-Hong Li, Feng-Ming You

**Affiliations:** Hospital of Chengdu University of Traditional Chinese Medicine, Chengdu 610072, China; laihengzhou@stu.cdutcm.edu.cn (H.-Z.L.); hjrclovercl@stu.cdutcm.edu.cn (J.-R.H.); fuxi884853@163.com (X.F.); ryftcm.dr@yahoo.com (Y.-F.R.)

**Keywords:** breast cancer, HER2-low, anti-HER2 therapies, antibody drug conjugates, novel combinations

## Abstract

**Simple Summary:**

HER2-low breast cancer (BC) accounts for more than half of breast cancer patients. Anti-HER2 therapy has been ineffective in HER2-low BC, for which palliative chemotherapy is the main treatment modality. The definitive efficacy of T-Dxd in HER2-low BC breaks previous treatment strategies, which will redefine HER2-low and thus reshape anti-HER2 therapy. This review summarizes detection technologies and novel agents for HER2-low BC, and explores their possible role in future clinics, to provide ideas for the diagnosis and treatment of HER2-low BC.

**Abstract:**

HER2-low breast cancer (BC) has a poor prognosis, making the development of more suitable treatment an unmet clinical need. While chemotherapy is the main method of treatment for HER2-low BC, not all patients benefit from it. Antineoplastic therapy without chemotherapy has shown promise in clinical trials and is being explored further. As quantitative detection techniques become more advanced, they assist in better defining the expression level of HER2 and in guiding the development of targeted therapies, which include directly targeting HER2 receptors on the cell surface, targeting HER2-related intracellular signaling pathways and targeting the immune microenvironment. A new anti-HER2 antibody-drug conjugate called T-DM1 has been successfully tested and found to be highly effective in clinical trials. With this progress, it could eventually be transformed from a disease without a defined therapeutic target into a disease with a defined therapeutic molecular target. Furthermore, efforts are being made to compare the sequencing and combination of chemotherapy, endocrine therapy, and HER2-targeted therapy to improve prognosis to customize the subtype of HER2 low expression precision treatment regimens. In this review, we summarize the current and upcoming treatment strategies, to achieve accurate management of HER2-low BC.

## 1. Introduction

HER2-low breast cancer (BC), defined as HER2 immunohistochemistry (IHC) 2+ and in situ hybridization (ISH)-negative or IHC1+, accounts for 40–50% of breast cancers [1,2]. Genomics analysis has shown HER2-low BC to be a heterogeneous disease [3,4]. While it is not known how the HER2-low subtype causes cancer, most patients have poor prognostic factors, such as larger tumor sizes, higher histological grades, and more regional lymph node involvement [2,5,6,7,8,9,10].

Denkert et al. investigated 2310 HER2-negative BC patients receiving neoadjuvant chemotherapy, and found that approximately 60% of hormone receptor (HR) positive BC patients also had low HER2 expression compared to 33% of triple negative breast cancer (TNBC) patients [11]. Compared to HER2-0 BC, HER2-low BC showed distinctive molecular features. The divergent rates of germline changes for BRCA1/2 and other BC susceptibility genes differed in both cases. Upon comparison of PIK3CA and TP53 mutations, it was particularly evident that HER2-0 and HER2-low BC have distinctly different genetic origins, which implies that HER2-lowBC has the potential to become a biological entity [11].

Unfortunately, the low HER2 expression failed to provide clinical prognosis benefits. The available HER2-targeted therapies are inefficient in HER2-low BC, and treatment options are restricted after the initial treatment progression [12,13]. When classified as HER2-negative and treated with palliative chemotherapy [1], even if HER2-0 has more adverse tumor characteristics than HER2-low, clinical trials also show no difference in survival outcomes between the two [11,14,15,16], raising the question of whether HER2-low BC is overtreated or undertreated?

Recent advances in drug development have, however, changed the current view, indicating that HER2-low BC may benefit from anti-HER2 therapy [17,18,19]. Moreover, in a longitudinal study, the inconsistent rate of HER2 expression in patients with advanced or recurrent BC was as high as 38%, and more than a third of HER2-0 BC patients exhibited a trend of transformation to HER2-low BC in metastatic lesions [20]. The possibility of improving the treatment of HER2-low BC is of great clinical significance.

This review addresses the detection technology used for HER2 expression and discusses novel agents for HER2-low BC, in order to evaluate dosing regimens through clinical trials and propose ideas for the diagnosis and treatment of HER2-low BC.

## 2. Detection and Diagnosis of HER2

Regarding the positive impact of antibody–drug conjugates (ADCs), the most urgent challenge is to refine the definition of low HER2 expression. Indeed, the core content of the American Society of Clinical Oncology/College of American Pathologists (ASCO/CAP) guidelines for HER2 testing recommends screening HER2 overexpressing populations that can benefit from trastuzumab, with a relatively vague description of the borderline range for low HER2 expression [21].

IHC/ISH is the only standard technique currently applied to define HER2 expression. IHC involves a reaction between antigens and antibodies, resulting in protein coloration of the receptor. Since the detection value shows a linear dynamic range, it is difficult to make an accurate comparison of inferior or superior quantitative results [22]. ISH utilizes nucleic acid molecular probes to examine HER2 gene amplification status, but the signals are subject to photobleaching and fading over time [23].

Given the limitations of IHC/ISH, a new method is needed to improve the concordance of HER2-low detection. In this context, IHC combined with other quantitative techniques can be used to determine the initial HER2 level before ISH is used to exclude gene amplification, and HER2 can then be defined in the form of continuous numerical variables [17,24]. Ultimately, the HER2 threshold for producing antitumor activity will need to be analyzed and clinical trials employed to broaden the population for which HER2-targeted therapy may be effective.

### 2.1. HERmark ™

HERmark™ (Monogram Biosciences Inc., South San Francisco, CA, USA) allows for the accurate detection of HER2 expression in FFPE tissue samples, covering the majority of the dynamic range from 0 to 3+ [25]. It was found to be highly concordant with conventional HER2 assays (IHC and ISH), but HERmark™ is highly sensitive and specific [26,27,28]. As a reliable quantitative assessment tool, HERmark™, can be used for HER2-low BC, particularly in the case of ambiguous IHC test results [25]. Nonetheless, the core technical requirements have restricted its application to the central lab, limiting its use in a broader range of clinical settings.

### 2.2. Real-Time PCR

Real-time PCR (RT-PCR) allows for rapid and quantitative gene amplification analysis, giving ISH-like results and accurate quantification of HER2 levels in nonoverexpressing samples [29,30]. OncotypeDX and MammaTyper, derived from RT-PCR, can predict the outcome of chemotherapy and the possibility of recurrence in early HR-positive BC by analyzing mRNA extracted from FFPE samples [31,32]. Unfortunately, false-negative PCR results are often found, principally due to differences in the distribution of cells between different types of tumors, the mixing of noninvasive cancer components in the process of DNA/RNA extraction and the destruction of mRNA integrity in FFPE samples [33,34,35].

### 2.3. Multiplex Ligation-Dependent Probe Amplification

A modified quantitative technique for PCR, multiplex ligation-dependent probe amplification (MLPA), allows for the analysis of multiple gene amplifications and different portions of gene deletions [36]. When there is a discrepancy between IHC and ISH, MLPA may identify HER2-low but ISH+ cases [37]. Similar to RT-PCR, MLPA may also lack the ability to detect tumor heterogeneity due to differences in sample cuts. Copy-number changes detected by MLPA should be verified with other methods [38].

### 2.4. Time Resolved Fluorescence Resonance Energy Transfer

Time-resolved fluorescence resonance energy transfer (TR-FRET) quantifies the fluorescence signals emitted from energy transfers between two adjacent molecules to assess HER2 expression [39]. With the application of long-life fluorophore in TR-FRET, a delay can be introduced between the excitation pulse and the signal measurement window, thus eliminating the short-lived background autofluorescence in FFPE materials. It thus has higher sensitivity and lower false-positive rate and false-negative rates [40].

The clinical effectiveness of anti-HER2 therapy is the only way to asses HER2 detection. To confirm clinical benefits, new technologies must be compared to baseline testing. Once HER2 expression has been identified, selecting the most suitable treatment is the next step.

Anti-HER2 agents are gaining traction in the treatment of HER2-low BC, indicating HER2 expression as a possible therapeutic breakthrough. In terms of the drug effect mechanism, therapies related to HER2 can be divided into those targeting HER2 receptors on the cell surface, those targeting HER2-related intracellular signaling pathways and those targeting the immune microenvironment, as shown in Figure 1.

## 3. Therapy

### 3.1. Targeting HER2 Receptors on The Cell Surface

HER2-mediated tumorigenic signal production can be blocked through the competitive binding of exogenous HER2 antibodies to HER2 on the cell membrane surface, thereby delaying tumor progression [41]. A summary of novel drugs is provided in Table 1.

#### 3.1.1. ADC

An ADC consists of an antibody against a target antigen, cytotoxic drugs (payload) and cleavable or non-cleavable linkers [42] (Figure 2). ADCs exert cytocidal effects through targeting of surface antigens, internalization, enzymatic cleavage and drug release [43], resulting in high target selectivity and potent lethality in chemotherapy.

Trastuzumab deruxtecan (T-DXd) is a broad-spectrum ADC composed of trastuzumab and a topoisomerase I inhibitor [44]. The payload of T-DXd proved to be highly membrane-permeable, exerting a powerful bystander effect [44]. Moreover, coupled with a drug structure different from that of the previously used paclitaxel and platinum-based chemotherapy for HER2-low BC, it resulted in a reduced risk of cross-resistance. As a corollary, it may become a good candidate for treatment of HER2-low BC.

The first clinical study (ClinicalTrials.gov identifier: NCT02564900) of T-DXd recruited 54 advanced HER2-low BC patients who progressed after standard treatment. Based upon the premise that the median number of treatment lines was up to 7.5, T-DXd performed spectacularly, with an objective remission rate (ORR) of 37.0% and a median duration of remission (DOR) of 10.4 months [45]. T-DXd was well-tolerated, with interstitial lung disease being the most prominent toxic reaction [46]. Excitingly, in the DESTINY-Breast04 phase III trial (ClinicalTrials.gov identifier: NCT03734029), significantly longer overall survival (OS) and progression-free survival (PFS) were seen in patients treated with T-DXd versus chemotherapy (OS, 23.9 vs. 17.9 months; hazard ratio (HR), 0.58; *p* = 0.001; PFS, 9.9 vs. 5.1 months; HR, 0.50; *p* < 0.001), leading to approval for T-Dxd in the NCCN and ASCO guidelines [47].

Additional efforts to broaden the applicability of T-DXd to a larger population, such as for HR+/HER2-low BC after the progress of endocrine therapy (ClinicalTrials.gov identifier: NCT04494425), are underway [48]. Moreover, combinations involving endocrine therapy, chemotherapy, immunotherapy and T-DXd are being explored (ClinicalTrials.gov identifier: NCT04556773) [49].

Other novel ADCs are RC48, SYD985 and A166, which use different cytotoxic drugs from T-Dxd (Table 2). They are currently in clinical development, showing encouraging results in phase I studies. In the C001CANCER phase I study (ClinicalTrials.gov identifier: NCT02881138) utilizing RC48, a significant improvement in OS and ORR was seen in an HER2-low cohort [50,51]. The SYD985.001 phase I study (ClinicalTrials.gov identifier: NCT02277717) evaluated all HER2-low BC patients who achieved a partial response (PR) with SYD985 [52]. In a phase I study (ClinicalTrials.gov identifier: NCT05311397), patients with relapsed or refractory solid cancers, including 51 with HER2-positive BC and 6 with HER2-low BC, received A166 [53]. To date, among the four evaluable patients with the HER2-low subtype, the disease control rate (DCR) for A166 was 75%, with manageable toxicity [54].

#### 3.1.2. Monoclonal Antibodies

Despite the ineffectiveness of trastuzumab for HER2-low BC, the ability of ADCs to significantly improve prognosis has spurred interest in conventional targeted agents.

MGAH22 is an Fc-engineered anti-HER2 antibody [58]. In vitro, MGAH22 has similar antitumor effects to trastuzumab, while additionally improving and enhancing its antibody-dependent cell-mediated cytotoxicity (ADCC) [58].

A phase Ⅱ trial of MGAH22 (ClinicalTrials.gov identifier: NCT01828021) in patients with relapsed or refractory advanced BC, including HER2-low BC, has completed enrollment. The final results will be announced in the near future [59].

#### 3.1.3. Bispecific Antibodies

HER2, an orphan receptor tyrosine kinase without a corresponding ligand, is often linked to EGFR, HER3 and HER4 [60]. Homogeneous or heterogeneous dimerization of HER2 rapidly activates downstream signaling cascades, thereby triggering tumor cell proliferation, migration, and invasion [61]. Bispecific antibodies (BsAbs) are single protein molecules that recognize two binding sites simultaneously, establishing an association between tumor cells and immune cells and blocking the appeal process [62].

A novel biparatopic antibody, MEDIA4276 binds to the 39S Fv and trastuzumab ScFV epitopes with site-specific conjugation to a microtubule inhibitor payload, inhibiting tumor cell proliferation more effectively than trastuzumab [63]. The antitumor effect of MEDI4276 was observed in several HER2-low cell lines in vitro. Further tests on HER2-low patient-derived xenografts (PDXs) excluded the interference of HR heterogeneity, which confirmed the tumor regression induced by MEDI4276 [64].

Unfortunately, in the phase Ⅱ dose-escalation and expansion study (ClinicalTrials.gov identifier: NCT02576548), even though MEDI4276 demonstrated obvious clinical activity, it still showed unbearable toxicity when the dose was higher than 0.3 mg/kg. Common toxicities included nausea, fatigue and vomiting [65]. The clinical development of MEDI4276 was severely delayed due to high incidences of drug-related adverse reactions, and it was therefore halted.

MCLA128 is a BsAb targeting HER2 and HER3 receptors with enhanced ADCC to directly inhibit tumor growth [66,67]. In 2017, researchers released the results of a phase Ⅱ study of MCLA128 (ClinicalTrials.gov identifier: NCT02912949). MCLA128 was administered with a median of 4.5 cycles to patients with HER2-positive metastatic BC who had received a median of 5.5 precious lines of metastatic therapy, and the clinical benefit rate (CBR) was 70% [68].

MCLA128 has the ability to overcome drug resistance in targeted therapy and endocrine therapy [66,69]. A phase Ⅱ study (ClinicalTrials.gov identifier: NCT03321981) enrolled patients with HER2-low BC who were estrogen receptor (ER)-positive and progressed to cyclin-dependent kinase 4 and 6 (CDK4/6) and endocrine therapy, with an effective 24 week CBR of 16.7% for MCLA128 combined with endocrine therapy [70]. No significant toxicity was observed.

#### 3.1.4. Trispecific Antibody

At the 2021 AACR Annual Meeting, representatives from the Sanofi organization presented a novel HER2-targeted T-cell splice agent called SAR443216. SAR443216 is a trispecific antibody that binds to HER2, CD3 and CD28 antigenic sites and contains mutant IgG4-Fc lacking an effector function [71]. The binding of CD28 can activate the IL-2 and NF-κB pathways, induce the anti-apoptosis protein Bcl-xL and, subsequently, enhance T-cell-dependent cytotoxicity (TDCC). In preclinical models, SAR443216 activated both CD4 and CD8 T cells in HER2-expressing cancer cell lines (including HER2-low), stimulated the secretion of cytokines and granzyme B, and exerted an antitumor effect [71].

The first phase I/IB open monotherapy trial (ClinicalTrials.gov identifier: NCT05013554) of TED16925 will recruit patients with different solid tumors expressing HER2, including those with HER2-low BC.

#### 3.1.5. Tyrosine Kinase Inhibitors

Tyrosine kinase inhibitors (TKIs) are pan-HER kinase inhibitors that interfere with or block HER2 signaling downstream by inhibiting tyrosine phosphorylation and the catalytic activity of the receptor [72]. As a result of its greater HER2 recognition ability and lower molecular weight, TKIs can prolong the action time of trastuzumab by regulating the ADCC effect and even cross the blood–brain barrier to protect against brain metastases [73].

Lapatinib, the first TKI approved against both HER2 and EGFR, enhanced HER2 expression in HER2-low BC cells [74], potentially transforming patients with refractory diseases into patients with tumors sensitive to trastuzumab [75,76,77,78].

Neratinib is an irreversible pan-HER inhibitor that more aggressively inhibits proliferation than lapatinib and may enable phenotypic alterations to increase ADCC mediated by trastuzumab [79,80,81,82,83].

Poziotinib, a novel oral quinazoline broad-spectrum HER inhibitor developed by Hanmi Pharm [84], overcomes the differences in drug binding sites, upregulates the expression of HER2 and enhances the activity of trastuzumab emtansine (T-DM1) [85]. In an open-label, multicenter, phase II clinical trial (ClinicalTrials.gov identifier: NCT02418689), a median PFS of 4.04 months was demonstrated when poziotinib was used in patients with refractory HER2-positive BC who had failed more than second-line HER2-targeted therapy [86,87]. Simmons et al. assessed efficacy outcomes gathered from eight clinical trials comparing third-line or higher therapy for HER2-positive BC. There were no differences in the results between T-Dxd and MCLA128, although poziotinib showed a survival benefit in a phase II trial [88].

Several other anti-HER2 TKIs, including pyrrolizidine, have been studied in light of these encouraging results (ClinicalTrials.gov identifier: NCT03412383) [89].

An issue that has emerged with TKI is the best application scenario for drugs. With the current results, TKI performs more as a synergist of HER2-targeting drugs, and related research is being explored.

### 3.2. Targeting HER2-Related Intracellular Signaling Pathways

#### Targeting PI3K/AKT/mTOR

The PI3K/AKT/mTOR signaling pathway is involved in the cell cycle, cell proliferation and angiogenesis and also regulates HER2 and ER receptor expression; it is thus a target for reversing endocrine resistance and HER2-targeted drug resistance [90,91,92]. Most studies combine PI3K inhibitors and Akt inhibitors with chemotherapy.

Allotype-specific PI3K inhibitors are widely used. They absorb the failure factors of pan-PI3K inhibitors; improve a variety of defects, such as off-target effects and toxicity; and specifically target the PI3Kp110 α, p110 β, p110 δ and p110 γ subtypes [93].

Alpelisib is the first PI3Kp110α isotype inhibitor to be approved by the FDA. Multiple clinical trials enrolling ER-positive BC patients have revealed the potential of alpelisib as a backline treatment option for endocrine-resistant BC [94,95,96,97,98]. The CBYL719XUS06T phase I/II study (ClinicalTrials.gov identifier: NCT02379247) investigated alpelisib with nab-paclitaxel in HER2-negative BC. In Phase I, 13 patients were divided into three groups treated with alpelisib (250, 300 and 350 mg) plus albumin paclitaxel, and no dose-limiting toxicity was observed [97]. In phase II, the median PFS was 8.7 months, and the ORR was 59%, with an overall response rate of 21% [99]. Regrettably, no further analysis was performed for the HER2-low subgroup.

AKT inhibitors are serine or threonine kinase molecules that target all AKT isoforms, strongly inhibit cell proliferation and AKT phosphorylation, and alleviate aggressive tumor behavior [100]. Almost all AKT inhibitors in clinical trials show limited therapeutic activities as single agents; therefore, combination therapeutic strategies are the main direction for research [101,102,103,104,105]. Nevertheless, conflicting results for AKT inhibitors in combination with chemotherapeutic agents have been observed in several clinical studies.

The BEECH phase Ib/II trial (ClinicalTrials.gov identifier: NCT01625286) evaluated capivasertib, an AKT inhibitor, with paclitaxel vs. paclitaxel alone in HR+/HER2- BC. There were similar dose intensities for paclitaxel between the treatment and control groups, with no significant difference in median PFS [106]. Comparatively, the PAKT trial (ClinicalTrials.gov identifier: NCT02423603) found capivasertib combined with paclitaxel to be superior to paclitaxel alone based on PFS (5.9 vs. 4.2 months; HR, 0.74; *p* = 0.06) [107,108].

In the FAIRLANE study, in which patients with TNBC received ipatasertib, another AKT inhibitor, with paclitaxel or paclitaxel alone, the pCR showed no clinical benefit or statistically significant improvement [109]. The PFS in LOTUS was improved under the same test conditions (ClinicalTrials.gov identifier: NCT02162719) [110].

In accordance with the different therapeutic effects of capivasertib or ipatasertib combined with chemotherapy, it is speculated that AKT mutations may not be the core driving event of cancer and that overactivation of the PI3K pathway cannot be effectively reduced by inhibiting AKT alone. Alternatively, a rapid attack may occur due to a tumor-acquired immune response or an inability to tolerate primary tumor drug resistance [111]. Since the pathways of cytotoxicity are unrelated to each other, AKT inhibitors should theoretically act synergistically with anti-HER2-targeted drugs.

### 3.3. Targeting the Immune Microenvironment

TNBC and HER2-positive groups present immunogenic features for BC, with a large number of tumor-infiltrating lymphocytes and higher levels of PD-L1, which are potential immunotherapy candidates [112,113]. Given the anti-tumor activity that HER2-targeted therapy can exert through the immune-mediated mechanism, immunotherapy and targeted therapy are being studied together.

#### 3.3.1. HER2-Derived Peptide Vaccine

Vaccinations are a form of active immunotherapy in which the immune system recognizes antigens on the surface of cells. Directly attacking tumor cells and tumor stroma or indirectly resetting the immune system to antitumor detection mode are the principles of action, which enhance the sustained effectiveness of the antitumor immune response [114]. Tumors can benefit from vaccination when conventional cytotoxic or targeted drug therapies fail [114].

Tumor-associated antigens (TAAs), including HER2, are the basis of many vaccines used for BC. Current immunogenic HER2-derived peptides derive from different parts of HER2 molecules, such as E75 from the extracellular domain, GP2 from the transmembrane domain and AE37 from the intracellular domain [115]. Their tumor killing effects are usually achieved by triggering the immune system to target HER2-expressing cells and induce a tumor-specific immune response [116].

The E75 peptide (nelipepimut-S, NP-S) is the most widely researched and advanced vaccine for BC. Several clinical studies have observed that the application of the E75 vaccine can induce an immune response and maintain safety [115,116,117,118]. A trial found that, due to the possible immune tolerance of HER2 positivity, patients carrying HER2-low were able to show a stronger immune response, suggesting the need for further clinical studies targeting the HER2-low subgroup.

Subsequently, the PRESENT trial (ClinicalTrials.gov identifier: NCT01479244) recruited lymph node-positive, low-to-moderate HER2-expressing early BC women, who were randomly assigned to granulocyte–macrophage colony-stimulating factor (GM-CSF) or NP-S combined with GM-CSF groups [119]. The interim report revealed that the trial did not demonstrate a clinical benefit for vaccination alone but instead was terminated early by an independent data monitoring committee due to a rapidly increasing number of recurrent events [120]. Paradoxically, vaccination with trastuzumab significantly prolonged DFS in clinical studies conducted simultaneously. In contrast, the DFS of the control group in the GP2 test was 89%. When the median follow-up was more than 34 months, the DFS was still 100%, which suggests that there may be a synergistic effect between the HER2-targeted peptide vaccine and trastuzumab [118,120].

A prospective, randomized, single-blind, placebo-controlled phase IIb study (ClinicalTrials.gov identifier: NCT01570036) was revalidated in light of the above results. In short, 275 patients were randomly assigned to receive NP-S or placebo after one year of standard treatment with trastuzumab [121]. 

Intention to treat (ITT) analysis was performed at a median follow-up of 25.7 months. No significant difference in DFS was observed in HER2 IHC 1+ or 2+ BC (HR, 0.62; 95% CI, 0.31–1.25; *p* = 0.18). However, the TNBC subset showed potential benefits [121]. Similar outcomes were seen again in the clinical trial for the AE37 vaccine [122,123]. To clarify the meaning of TNBC in the context of HER2-derived peptide vaccines, further discussion should be given to HR status and HER2-0 and HER2-low subgroups.

#### 3.3.2. Immune Checkpoint Inhibitor

Immune escape is the key mechanism of tumor occurrence and development. Generally, tumors block T-cell activation by connecting immune checkpoint receptors (ICR) with their ligands. A local microenvironment containing inflammatory cytokines can also induce the undifferentiated expression of PD-1 [124]. In preclinical experiments, blocking the inhibitory pathway of ICR–ligand interactions in the tumor microenvironment restores the functional immune response of TAAs [125,126]. Several monoclonal antibodies targeting ICR markers, including cytotoxic T lymphocyte-associated antigen (CTLA-4), programmed cell death protein 1 (PD-1) and programmed death ligand 1 (PD-L1), have shown good clinical activity in BC [127,128,129,130].

Trials are being conducted to determine whether immune checkpoint inhibitor (ICB) is effective when used in combination with anti-HER2 therapies. In a dose-escalation test of DS8201-A-U105 (ClinicalTrials.gov identifier: NCT03523572), the recommended dose (RDE) of T-Dxd combined with nivolumab was 5.4 mg/kg and 360 mg, respectively [131].

For the HER2-low cohort (n = 16), in the phase II dose-expansion trial, RDE was administered to 48 participants with an ORR of 37.5% and median PFS of 6.3 months (95% CI, 2.3-NE) after approximately 7 months of follow-up [132]. Despite the fact that T-Dxd combined with navumab showed an ORR similar to T-Dxd monotherapy, given that combination therapy has a much higher benefit rate than any single drug in preclinical models, clinical benefits cannot be determined without a longer follow-up.

Duvalizumab is a selective and high-affinity monoclonal antibody against human immunoglobulin G1K that completes the process of T-cell recognition and tumor cell killing by blocking the binding of PD-L1 to PD-1 and CD80 [133].

Cohort 6 of the multicenter, randomized, IB/II BEGONIA study (ClinicalTrials.gov identifier: NCT03742102) was designed to evaluate the potency of dovalizumab with T-DXD as a first-line treatment for HER2-low BC patients [134]. Eighteen patients had received more than one treatment at the time of the report, and data for twelve identifiable case assessments were obtained. Treatment resulted in 8 SDs and 1 PR, with an ORR of 66.7% [134]. Despite limited patient numbers, the combination showed positive safety and effectiveness.

Another study of pembrolizumab in combination with T-Dxd (ClinicalTrials.gov identifier: NCT04042701) is underway in light of the favorable efficacy of PD-1 inhibitors in HER2-low BC. The second phase of the trial will give the HER2-low cohort RDE treatment to obtain ORR outcomes through independent central evaluation (ICR), with preliminary results expected in May 2023 [135].

## 4. Other Therapies

Although there is more enthusiasm to develop tumor biologic therapies, given that chemotherapy and endocrine therapy are the main treatments for BC, it is necessary to compare the advantages and disadvantages with targeted therapies and determine the best order of sequential use.

### 4.1. Endocrine Therapy

In a retrospective study involving 3689 BC patients examining the intrinsic subtype distribution of PAM50, the HER2-low BC population was approximately 80.8% HR+. It is therefore crucial to examine whether HER2 expression levels impact the efficacy of endocrine therapies [1,10,136].

In the BIG1-98 trial, 3650 postmenopausal women were given letrozole and tamoxifen. Analysis of heterogeneity demonstrated that HER2 expression levels were not affected by the drug or the effects of endocrine therapy [137]. The TRANS-AIOG meta-analysis integrated data from three studies, ATAC, BIG1-98 and TEAM, and compared therapeutic strategies based on letrozole, tamoxifen or switching from letrozole to tamoxifen and found evidence, supporting the view that endocrine therapy can benefit various levels of HER2 expression [138]. AglaiaSchiza et al. further measured the effect of HER2-targeted therapy and denied the predictive value of HER2 status in postmenopausal BC patients for endocrine therapy. They reiterated that endocrine therapy is still the first option for HR-positive BC regardless of HER2 expression [139,140].

With the advent of endocrine therapy, drug resistance has increased. CDK4/6 inhibitors (CDK4/6i) are considered effective for retaining endocrine sensitivity by interfering with the ER cascade, inhibiting RB1 phosphorylation, and triggering G1 to S cell cycle arrest [141,142]. Since 2014, the MONARCH, PALOMA, and MONALEESA studies have demonstrated that CDK4/6i is gradually changing the course of endocrine therapy for advanced BC [143,144,145,146,147,148,149]. Palbociclib, ribociclib, and abemaciclib are currently approved as first-line treatments for HR+/HER2-negative BC in combination with aromatase inhibitors (AI) or as second-line treatments in combination with fulvestrant.

In addition, some scholars believe that bidirectional crosstalk between members of the human epidermal growth factor receptor family (HER) and the estrogen receptor (ER) is the basis of drug resistance; that is, high expression and amplification of HER2 drive the occurrence of endocrine resistance [150]. Kelvin et al. investigated the relationship between HER2 expression levels and the efficacy of CDK4/6i combined with letrozole or fulvestrant therapy in ER+/HER2-negative BC patients [151]. In HER2-low BC, the PFS was significantly shorter than in HER2-0 (8.9 months vs. 18.8 months, *p* = 0.014) [151]. HER2 upregulation is responsible for endocrine resistance, resulting in reduced PFS and pCR rates with overall endocrine therapy.

The ER and HER axes are being targeted simultaneously in order to improve the response of the HR+/HER2+ population to endocrine therapies and control the onset of endocrine resistance. Triple therapies targeting HER2, HER3 and ER evaluated in preclinical studies were effective in the ER+/HER2-low BC pdx model. Lumretuzumab (anti-HER3) and patuximab (anti-HER2) combined with fulvestrant maintained durable antitumor effects [69]. This drug combination has not shown promising safety or antitumor activity in clinical trials, and it has been limited by DLT, has a narrow treatment window and shows a high incidence of diarrhea [152].

Excitingly, the triple combination has been reversed with the development of the bispecific antibody MCLA-128, which directly and simultaneously targets HER2/HER3 and also greatly reduces drug toxicity [153]. In a phase II trial recruiting patients who had progressed after treatment with ET and CDK4/6i, adding MCLA-128 to the ET backline resulted in clinical benefit and even reversed endocrine sensitivity in 17% of these patients [70].

The NA-PHER2 trial (ClinicalTrials.gov identifier: NCT02530424) evaluated the drug combination of palbociclib, fulvestrant and dual HER2 blockade (trastuzumab, pertuzumab) [154]. The HR+/HER2-low BC cohort performed well at the endpoint, with Ki67 decreasing consistently from baseline to 2 weeks after treatment and before surgery (16 weeks) [155]. The above results show the preliminary efficacy of endocrine therapy involving CDK4/6i in combination with HER2-targeted drugs for HR+/HER2-low BC [155,156]. They confirm direct crosstalk between the HER and ER axis, and any single targeted therapy has limitations. HER-targeted therapy and endocrine therapy are therefore likely to be effective for patients with ER+/HER2-low BC who are resistant to endocrine therapy.

### 4.2. Chemotherapy

From the perspective of chemotherapy benefit, several clinical trials have found that HER2-low BC has a lower pCR rate than HER2-0 BC. However, the difference is not statistically significant, and the OS and DFS prognostic outcomes remain uncertain [157,158,159]. The currently available evidence suggests that chemotherapy regimens for HER2-low BC can still be managed with reference to HER2-negative BC. The difference is that paclitaxel- and anthracycline-based regimens for chemotherapy are preferred for the HR-negative subgroup (TNBC) in HER2-negative BC, regardless of early and late stages, whereas endocrine therapy is generally preferred in the HR-positive subgroup (luminal A/luminal B). Additional sequential or combination chemotherapy is recommended for intermediate/high-risk patients. Chemotherapy is recommended as the first line of treatment when the disease is critical and, progresses rapidly, and the ER is low [160,161,162].

## 5. Discussion and Future Prospects

Several new drugs now have shown clinical evidence that they can be used to treat HER2-low BC as part of HER2-targeted therapy, reflecting the clearly unmet treatment needs of HER2-low BC.

Researchers are exploring targeted therapies for HER2-low BC, but HER2 expression levels must be identified to guide treatment. While IHC/ISH assays have become an accepted standard of determination, their accuracy is low due to technical shortcomings. Other quantitative analysis techniques offer advantages and disadvantages, but they are limited by the core technical requirements, and cannot enter large-scale clinical trials; thus far, a unified standard that is suitable for use as a verification tool for IHC/ISH has not been formed.

Different HER2 expression levels respond differently to novel HER2-targeted therapies. The threshold for HER2 grouping should be redefined, abandoning the traditional dichotomy of HER2-positive and -negative. By repeatedly measuring HER2 expression levels across different disease nodes, we can redefine the treatment potential and molecular typing of HER2-negative breast cancer.

As a backbone therapy for patients with HER2-low BC, ADC analogs are unquestionably effective, but their efficacy against early-stage disease remains to be further confirmed. On the basis of the available evidence, we recommend that HER2-low BC be treated as two ER+/ER- groups.

Early-stage tumors are still managed as HER2-negative tumors. Patients carrying an ER+ will likely benefit from hormone therapy, and those with an ER- may benefit from anthracycline- or paclitaxel-based chemotherapy. In cases where the tumor is at an advanced stage, T-DXd can be applied first. When the results from inhibiting a single pathway are inadequate, further chemotherapy, endocrine therapy or immunotherapy can be considered. Clinical trials are prioritized whenever resistance occurs in the backline. Since there are a limited number of clinical trials stratified for HER2-low BC, the above results are still susceptible to error. There will, however, be an increasing number of studies that will focus on HER2-low BC in the future, providing an empirical basis for anti-HER2 therapeutics and creating more survival opportunities for HER2-low BC patients.

## Figures and Tables

**Figure 1 cancers-14-03774-f001:**
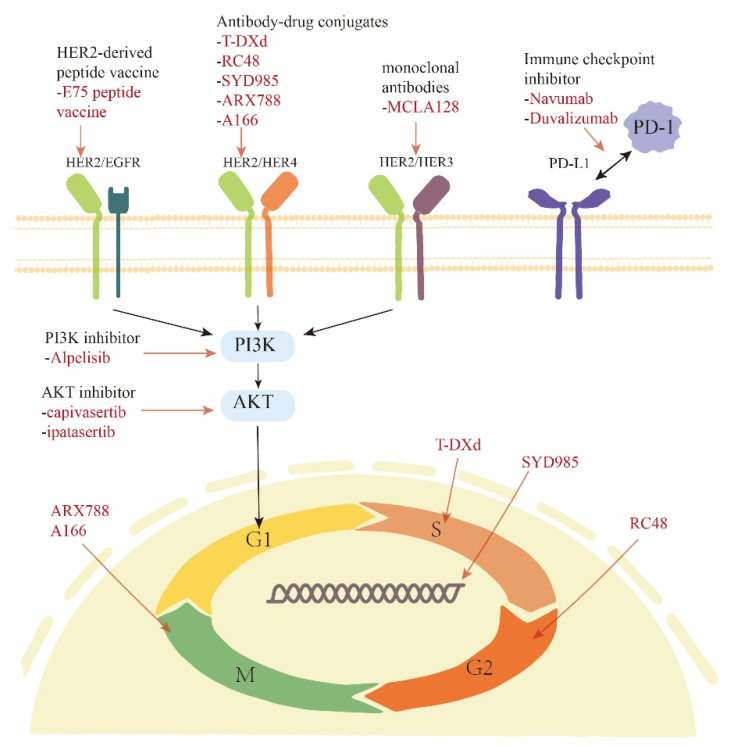
HER2-related therapies for HER2-low breast cancer.

**Figure 2 cancers-14-03774-f002:**
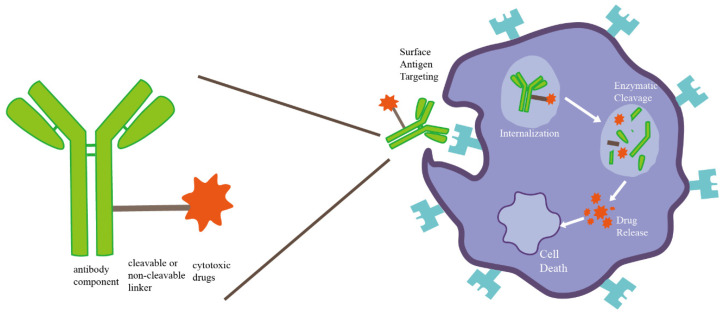
Schematic diagram of ADC composition structure and drug mechanism.

**Table 1 cancers-14-03774-t001:** Summary of novel drugs targeted HER2 receptors in development for HER2-low BC.

Drug	Target	Representative Clinical Trial	Patient Cohort	N	Treatment Arms	Main Efficacy Results	Toxity
T-DXd	HER2	DS8201-A-J101 trial NCT02564900	Metastatic HER2-low BC	54	T-DXd	ORR: 37.0%	Interstitial lung disease, anemia, diarrhea
RC48	HER2	C001CANCER phase I trial NCT02881138	Advanced malignant solid tumors with HER2+	118 (HER2-low BC: 48)	RC48	ORR: 39.6%; mPFS: 5.7 months	Hypoesthesia, fatigue
SYD985	HER2	SYD985.001 phase I trial NCT02277717	Advanced BC or gastric, urothelial, or endometrial cancer with at least HER2 IHC 1+	146 (HER2-low BC: 47)	SYD985	ORR in HR+/HER2-low BC: 28%; ORR in HR-/HER2-low BC: 40%	Atigue, conjunctivitis, dry eye
A166	HER2	KL166-I-01-CTP NCT05311397	Solid tumors with HER2 expression	57 (HER2-low BC: 6)	A166	DCR: 75%	Keratitis, decreased appetite, dry eye, vision blurred
MEDI4276	HER2	D5760C00001 NCT02576548	HER2 expressing BC or gastric/stomach cancers	47	MEDI4276	NA	Nausea, fatigue, vomiting
MCLA128	HER2, HER3	MCLA-128-CL02 NCT03321981	Metastatic BC	106 (HER2-low BC: 48)	MCLA128 with ET	DCR: 45%	Fatigue, diarrhea, nausea
SAR443216	HER2	TED16925 trial NCT05013554	Relapsed/refractory HER2 expressing solid tumors	NA	SAR443216	NA	NA

**Table 2 cancers-14-03774-t002:** Structural characteristics of ADCs in ongoing clinical trials.

ADC	HER2-Targeting Antibody	Linker	Cytotoxic Drug	Ongoing Clinical Trials with HER2-low BC
T-DXd	Trastuzumab	Cleavable	Topoisomerase I inhibitor	NCT04494425 NCT04556773
RC48 [55]	Hertuzumab (anti-HER2 humanized Ab)	Cleavable	MMAE	NCT04400695 NCT04965519
SYD985 [56]	Trastuzumab	Cleavable	Duocarmycin analogs	NCT04205630 NCT04602117 NCT04235101
A166 [57]	Trastuzumab	Cleavable	Microtubule inhibitor	NCT03602079

ADC, antibody-drug conjugate; BC, breast cancer; T-DXd, trastuzumab deruxtecan; Ab, antibody; MMAE, monomethyl auristatin E.

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
