# Peer review of "Targeted Approaches to HER2-Low Breast Cancer: Current Practice and Future Directions"

_cancers, 2022, doi:10.3390/cancers14153774_

Round 1

Reviewer 1 Report

The paper is a comprehensive review on treatment modalities of Her2-low breast cancer. 

The article should be named review...

The most interesting part would be to properly diagnose the patients who benefit from the new approaches. therefore the authors should expand and deepen the diagnostic sections to bring the reader up-to date regarding new improvements.

The Destiny-Breast 04 Study was published in the NEJM  and should be discussed. 

There are some minor spell checks necessary.

Author Response

Response to Reviewer 1 Comments

Thank you for sending the reviewers’ comments on our manuscript entitled “Targeted approaches to HER2-low breast cancer: current practice and future directions” (cancers-1827409). We have studied the comments carefully and have made corrections which we hope it will meet with approval. All the changes are marked in red in the revised manuscript.

The answers to the reviewer’s comments are as follows:

Comments: The paper is a comprehensive review on treatment modalities of Her2-low breast cancer.

Response: We are very appreciated the reviewer’s insightful comments. In this version, we have polished the English writing. And we reorganized some analysis, figures and description. We hope it could satisfy this journal with high level. Due to some modification, the figures number and line number have changed in the revised manuscript.

Point 1: The article should be named review...

Response 1: Thanks for the reminding. We have checked all statements about article types to avoid such problems.

Point 2: The most interesting part would be to properly diagnose the patients who benefit from the new approaches. therefore the authors should expand and deepen the diagnostic sections to bring the reader up-to date regarding new improvements.

Response 2: Thank you for the detailed review. For this question, we have revisited the detection methods means proposed in this paper and added the sentence of “1.4. Time Resolved Fluorescence Resonance Energy Transfer” to the original content.

Indeed, this paper briefly introduces several novel diagnostic methods of HER2 expression except IHC and ISH, which is intended to show that by combining the existing technology, the shortcomings of IHC/ISH can be compensated for in terms of, for example, sample cut differences, fluorescence signal variations, and a wide range of data thresholds, improving the consistency of IHC/ISH detection and accurately defining the HER2 low expression BC population. However, since the core of this article is the systemic treatment of HER2-low breast cancer, it is not appropriate to expand too much space to review diagnostic techniques.

Point 3: The Destiny-Breast 04 Study was published in the NEJM and should be discussed.

Response 3: Thanks for your great suggestion on improving the accessibility of our manuscript. We fully agree with your comment. The new sentence is now written as “Excitingly, in the DESTINY-Breast04 phase III trial (ClinicalTrials.gov identifier: NCT03734029), a significantly longer overall survival (OS) and progression-free sur-vival (PFS) was seen in patients treated with T-DXd, versus chemotherapy (OS, 23.9 vs. 17.9 months; hazard ratio [HR], 0.58; P=0.001; PFS, 9.9 vs. 5.1 months; HR, 0.50; P<0.001), leading to NCCN and ASCO guidelines approval T-Dxd [48].”

Point 4: There are some minor spell checks necessary.

Response 4: Thank you for your suggestion. Corrections were made to the spelling mistakes in the paper after careful review.

We would like to take this opportunity to thank you for all your time involved and this great opportunity for us to improve the manuscript. We hope you will find this revised version satisfactory.

Sincerely,

The Authors

Reviewer 2 Report

The article is very insightful and necessary for tackling a very important patient group. However below are my comment.

1. Line 40 - The authors should use the intro to set the premise for the study or review. More information is needed about how is HERlow over treated or under treated as that can also be the motivating factor.

2. Line 93 - Introduction missing

3. Line 94 - Please add the full form of ADC

4. Line 93 onwards - The formatting makes it very hard to read th actual text. Please put the headings in ways that is easy for the readers.

5. Line 96 - This introduction doesnt shed any light on the structural orientation of ADC. Also a corresponding pictorial representation would be nice.

6. Line 99 - Does the 42 have any meaning?

7. Line 109 - This statement seems to be without context - what kind of study are the authors referring to here for getting info about the effectiveness of this ADC.

8. Line 164 - Recommendation - Rather than having each of the ADCs described the authors can club all of them together in one paragraph and keep referring the table for the granular details about the ADCs.

9. Line 247 - The authors have provided a lot of info about the different drugs but have digressed from the topic of HER2 low BC

Author Response

Response to Reviewer 2 Comments

Thank you for sending the reviewers’ comments on our manuscript entitled “Targeted approaches to HER2-low breast cancer: current practice and future directions” (cancers-1827409). We have studied the comments carefully and have made corrections which we hope it will meet with approval. All the changes are marked in red in the revised manuscript. The answers to the reviewer’s comments are as follows:

Comments: The article is very insightful and necessary for tackling a very important patient group. However below are my comment.

Response: We are very appreciated the reviewer’s clear and detailed feedback and hope that the explanation has fully addressed all of your concerns. In the remainder of this letter, we discuss each of your comments individually along with our corresponding responses.

Point 1: Line 40 - The authors should use the intro to set the premise for the study or review. More information is needed about how is HERlow over treated or under treated as that can also be the motivating factor.

Response 1: Thanks for the reminding. In the article, we corrected it as following:

Line 31: Unfortunately, the low HER2 expression failed to provide clinical prognosis bene-fits. The available HER2-targeted therapies are inefficient in HER2-low BC, treatment options are restricted after the initial treatment progression [12,13]. When classified as HER2-negative and treated with palliative chemotherapy [1], even if HER2-0 has more adverse tumor characteristics than HER2-low, clinical trials also show no difference in survival outcomes between the two [11,14-16], raising the question of whether HER2-low BC is overtreated or undertreated?

Point 2: Line 93 - Introduction missing

Response 2: Thank you for your suggestion. For this question, we have added the sentence of “Blocking HER2-mediated tumorigenic signal production by exogenous HER2 antibodies binding competitively to HER2 on the cell membrane surface, thereby delaying tumor progression.” in the Line 111.

Point 3: Line 94 - Please add the full form of ADC

Response 3: Thank you for the detailed review. We have written the full form of ADC in the Line 48.

Point 4: Line 93 onwards - The formatting makes it very hard to read the actual text. Please put the headings in ways that is easy for the readers.

Response 4: Thanks for reminding me. Corrections were made to the format in the paper after careful review.

Point 5: Line 96 - This introduction doesnt shed any light on the structural orientation of ADC. Also a corresponding pictorial representation would be nice.

Response 5: Thank you for your suggestion. In the article, we added it as following: An ADC consists of the antibody against target antigen, cytotoxic drugs (payload) and cleavable or non-cleavable linkers [43] (Figure 2). It exerts cytocidal effects through surface antigen targeting, internalization, enzymatic cleavage and drug release [44], raising a high target selectivity and the potent lethality of chemotherapy.

Figure 2. Schematic diagram of ADC composition structure and drug mechanism. (Please see the attachment)

Point 6: Line 99 - Does the 42 have any meaning?

Response 6: Thanks for reminding me. We apologize that this is a spelling mistake. In the article, we corrected it as following:

Line 120: Trastuzumab deruxtecan (T-DXd) is a broad-spectrum ADC composed of trastuzumab and a topoisomerase I inhibitor [45].

Point 7: This statement seems to be without context - what kind of study are the authors referring to here for getting info about the effectiveness of this ADC.

Response 7: Thank you for the detailed review. In this section, a number of international multicenter clinical studies of T-DXd are under way to determine its safety and efficacy in the treatment of HER2-lowBC, rather than that T-DXd has been proven to be effective. This phrase has been removed to avoid misunderstanding.

In addition, T-DXd has made great progress according to the DESTINY-Breast04 phase III trial. In the Line 130, I added the sentence of “Excitingly, in the DESTINY-Breast04 phase III trial (ClinicalTrials.gov identifier: NCT03734029), a significantly longer overall survival (OS) and progression-free sur-vival (PFS) was seen in patients treated with T-DXd, versus chemotherapy (OS, 23.9 vs. 17.9 months; hazard ratio [HR], 0.58; P=0.001; PFS, 9.9 vs. 5.1 months; HR, 0.50; P<0.001), leading to NCCN and ASCO guidelines approval T-Dxd [48].”

Point 8: Line 164 - Recommendation - Rather than having each of the ADCs described the authors can club all of them together in one paragraph and keep referring the table for the granular details about the ADCs.

Response 8: Thanks for suggestion. In the article, we corrected it as following:

Line 141: Other novel ADCs are RC48, SYD985 and A166, which have different cytotoxic drugs from T-Dxd (Table 2). They are currently in clinical development, showing encouraging results in phase I studies. In the C001CANCER phase I study (ClinicalTri-als.gov identifier: NCT02881138) treated with RC48, a significant improvement in OS and ORR was seen in HER2-low cohort [54,55]. The SYD985.001 phase I study (Clini-calTrials.gov identifier: NCT02277717) evaluated all HER2-low BC patients who achieved a partial response (PR) with SYD985 [56]. In a phase I study (ClinicalTri-als.gov identifier: NCT05311397), patients with relapsed or refractory solid cancers, including 51 with HER2-positive BC and 6 with HER2-low BC, received A166 [57]. To date, among 4 evaluable patients with the HER2-low subtype, disease control rate (DCR) to A166 was 75%, with manageable toxicity [58].

Table 2. Structural characteristics of ADCs in ongoing clinical trials.

ADC

HER2 Targeting Antibody

Linker

Cytotoxic Drug

Ongoing Clinical Trials In HER2-low BC

T-DXd

Trastuzumab

Cleavable

Topoisomerase I Inhibitor

NCT04494425

NCT04556773

RC48 [51]

Hertuzumab (anti-HER2 humanized Ab)

Cleavable

MMAE

NCT04400695

NCT04965519

SYD985 [52]

Trastuzumab

Cleavable

Duocarmycin Analogs

NCT04205630

NCT04602117

NCT04235101

A166 [53]

Trastuzumab

Cleavable

Microtubule Inhibitor

NCT03602079

ADC, antibody drug conjugate; BC, breast cancer; T-DXd, trastuzumab deruxtecan; Ab, antibody; MMAE, monomethyl auristatin E.

Point 9: Line 247 - The authors have provided a lot of info about the different drugs but have digressed from the topic of HER2 low BC

Response 9: Thanks for your great suggestion on improving the accessibility of our manuscript. In this paper, we review the drugs used to treat HER2-lowBC and evaluate the dosing regimens through clinical trials. Whether targeted therapy, immunotherapy, endocrine therapy and chemotherapy are monotherapy or combination regimens, the purpose is to evaluate the timing of drug application and explore the best treatment for HER2-low BC in different periods.
